# Effect of Different Water Salinity Levels on the Germination of Imazamox-Resistant and Sensitive Weedy Rice and Cultivated Rice

**Silvia Fogliatto ***, **Francesca Serra, Lorenzo Patrucco, Marco Milan** and **Francesco Vidotto**

Department of Agricultural, Forest and Food Sciences, University of Turin, I-10095 Grugliasco, Italy;
francesca.serra@unito.it (F.S.); lorenzo.patrucco@unito.it (L.P.); marco.milan@unito.it (M.M.);
francesco.vidotto@unito.it (F.V.)
* Correspondence: silvia.fogliatto@unito.it; Tel.: +39-0116708897

**Abstract:** Weeds that have become resistant to herbicides may threaten rice production. Rice cultivation is mainly carried out in coastal and river delta areas that often suffer salinity problems. The aims of this study were to evaluate the effects of salinity upon germination and the root and shoot seedling growth of Italian weedy rice and cultivated rice (*Oryza sativa*), and to find a possible correlation between salinity and herbicide resistance. Seed germination tests were conducted in Petri dishes on four imazamox-sensitive and one resistant weedy rice populations and two rice varieties: Baldo (conventional) and CL80 (imidazolinone-resistant Clearfield® variety). Different salt concentrations were tested: 0, 50, 100, 150, 200, 250, 300, 350 and 400 mM NaCl. Germination percentage, germination speed, seedling root and shoot length were affected by increasing the salt concentration in all tested populations and varieties. The germination percentage was in general more affected in resistant weedy rice and CL80. In resistant weedy rice this was partially compensated by a faster germination up to 100 mM. In terms of seedling root and shoot length, CL80 and Baldo showed the highest tolerance to salt; resistant weedy rice was not able to produce seedling roots and shoots at concentrations > 300 mM.

**Keywords:** red rice; *Oryza sativa*; salt; herbicide resistance

---

## 1. Introduction

Rice is one of the most cultivated crops, with more than two billion people depending on it as their main source of calories [1]. Rice quality and yield are fundamental for the whole world, especially considering population rise, expected to be from 9 to 10 billion by 2050 [2]. One of the main obstacles for crop production is represented by weeds, which are considered one of the main causes of yield reduction [3]. In rice cultivation, weedy rice (*Oryza sativa* L.) is one of the most troublesome weeds worldwide, able to cause severe yield losses, mainly because of its high competitiveness [4–6].

The control of weedy rice is problematic in rice cultivation due to its botanical similarity to the crop, as they belong to the same species and, as a consequence, the herbicides applied to control the weed can cause serious damage also to the rice [7–10].

The introduction of imidazolinone-resistant rice varieties (IMI rice), known as Clearfield®, has offered to farmers an efficient tool to control weedy rice in post-emergence selectively [8]. The Clearfield® technology has developed rice varieties, through classic breeding methods, which are characterised by resistance to the imidazolinone herbicides (imazamox in Europe) [10], which belong to the group of acetolactate synthase (ALS)-inhibitors. The repeated use of imidazolinone herbicides and the cross pollination between rice and its weedy relatives has caused the diffusion of resistant populations of weedy rice, both in North and South America and Europe [11–13].

Several weedy rice populations resistant to imazamox were identified only four years (in 2010) after the introduction of Clearfield® to Italy, which occurred in 2006 [8,14].

The efficacy of Clearfield® technology is still quite high for weedy rice and several other species, even if new cases of resistance are reported every year. Moreover, the problem of resistance is worsened by the fact that the majority of herbicides used in Italian rice cultivation are ALS-inhibitors. Different weed species have developed resistant populations, such as *Echinochloa crus-galli*, *Cyperus difformis*, *Schoenoplectus mucronatus*, *Alisma plantago-aquatica*, *Ammania coccinea*, and weedy rice [15]. Cases of resistance in these species are reported for the main Italian rice areas, in particular across the Piemonte and Lombardia regions and in the Po delta [15].

Besides herbicide resistance, weedy rice is also able to adapt quickly to various environments because of its high polymorphism that permits the identification of several populations that differ for their morphological characteristics, such as plant height, hull colouration, awn presence, or variability in some physiological traits, including a variable degree of dormancy, time of flowering, tolerance to different abiotic stresses, etc. [16].

Some of the most important abiotic stresses that many rice cultivation areas have to face worldwide are soil and water salinisation [17]. According to the United Nations Environment Programme (UNEP), 20% of the agricultural land and 50% of the cropland worldwide suffer from salinity [18]. Soil salinisation is favoured by many factors, such as the improper planning of crop irrigation, high evapotranspiration and, in coastal areas, saltwater intrusion and excessive water exploitation [19,20].

Salinisation and saltwater intrusion are a serious threat to rice cultivation, and to agriculture in general, in coastal areas of tropical regions [21]. However, some rice areas in Europe are also affected by the same problem, in particular the Mediterranean countries in which rice is cultivated, namely Spain, Portugal, Italy, Greece and France [22,23]. In Italy, in which is located the majority of the European rice area, this phenomenon is more problematic in Sicily and Sardinia, in the Tiber delta, in Versilia and in the Adriatic coastal area, especially in the Po plain [24,25].

The effects of salinity on crops are multiple, such as the reduction of growth due to high osmotic potential and ion toxicity, photosynthesis inhibition by decreased $CO_2$ availability and pigment content, diminished nutrient and water uptake, and root damage; all of these phenomena eventually result in yield reduction [21,26]. One of the crop growth stages most affected by salinity is seed germination, a phase in which plants are generally quite sensitive to salt stress [27]. Moderate levels of salinity cause an increase of abscisic acid (ABA) production, which is known to induce or maintain seed dormancy, which results in a reduction in the germination level and a delay in germination time, especially in halophyte species [27,28]. High rates of salinity are also responsible for stronger hormonal alterations and other effects, such as the reduction in water uptake and cell damage that reduce the embryo viability and hamper the germination in sensitive species, or postpone the start of germination in tolerant species [27,29,30]. In rice, previous studies have shown that salinity acts mainly on delaying seed germination than reducing germination percentage; moreover, salt tolerance in rice increases with time, being lowest during the early seedling stage (2–3 leaf stage) with a certain variability amongst different rice varieties [31,32]. A previous study found that, beside the variability among varieties, only few were slightly affected by a salinity level of 12 dS m$^{-1}$ at the seedling stage [32]. The salinity level tolerated by rice during germination seems not to be correlated with salinity tolerance at other growth stages [33].

Nevertheless, salinity has effects, not only on the crop but also on weeds; even though the response of crops to salinity stress has been largely investigated, knowledge about the potential impact of salinity upon the weed community is more limited. Moreover, few studies evaluated, even for rice, the effects of salt at the germination stage. The few available studies conducted on weedy rice and salt stress highlighted a moderate tolerance of the weed, similar to that of *Echinochloa crus-galli* (L.) P. Beauv. and higher compared to *Cyperus difformis* L. [34].

Information on the possible correlation between salinity response and herbicide resistance on rice varieties and weedy rice is not available, even though this is of great importance to understand if herbicide resistance can be a more serious problem in saline areas.

Thus, the present studies had the aim of evaluating the effect of salinity on the germination of some Italian weedy rice populations and cultivated rice varieties; moreover, the presence of a potential relationship between the response to salinity and sensitivity to imazamox were investigated.

## 2. Materials and Methods

The effect of water salinity on rice and weedy rice germination was evaluated by carrying out a series of germination trials. In 2015, five weedy rice populations having different morphological traits (awn presence, plant height) were chosen among a larger set of about 150 populations to be representative of the variability of the Italian weedy rice populations; these were collected in 2008–2009. All of the weedy rice populations came from the rice area across Piemonte and Lombardia. From previous greenhouse tests, four populations (labelled s1, s2, s3 and s4) were sensitive to the label rate of imazamox, while one (labelled r) was resistant. The resistance of the r population was further confirmed with a genetic test that showed the presence of a target site mutation (amino acid substitution: Ser653Asp). In addition, the rice varieties Baldo (a conventional variety), and CL80 (Clearfield® variety resistant to imazamox), were used in the study. Baldo was chosen, not only as being not resistant to imazamox, but also because it was one of the most cultivated rice varieties in Italy, with an economic importance also for the export market [35]. This variety has demonstrated a certain degree of salt tolerance at the seedling stage [36], but only poor information is available on the germination response to salt. The variety CL80 was included as representative of imazamox-resistant varieties.

Weedy rice seeds were air dried for one week after collection and then stored in the refrigerator at about 4 °C until used in this experiment.

In order to test salt stress resistance, 20 seeds of each population were placed in a 9 cm Petri dish and lined with a filter paper imbibed with either 5 mL of deionised water (control) or with the same amount of different saline solutions; seeds were placed in Petri dishes working under a laminar flow cabinet and sealed with Parafilm to avoid drying and contamination. Nine different salt concentrations were prepared by dissolving NaCl in deionised water in concentrations equal to 0 mM (control), 50 mM, 100 mM, 150 mM, 200 mM, 250 mM, 300 mM, 350 mM and 400mM.

The salt concentrations were chosen on the basis of similar previous studies and on the salinity levels recorded in selected European rice cultivation areas [13,37–39]. Petri dishes were incubated in a growth chamber at 25 °C with alternating light/dark (16/8 h) at the Department of Agricultural, Forest and Food Sciences of the University of Turin. All treatments were arranged in a completely randomised design with three replicates; a single Petri dish was considered as one experimental unit.

Every day for fifteen days, the Petri dishes were examined in order to count the germinated seeds. At the end of the germination test, the seed germination percentage and root and shoot length of 10 seeds randomly chosen in each Petri dish were measured. All the ungerminated seeds were considered viable, as they were all firm when pressed with tweezers, and no seeds were soft, discoloured, or mouldy [40,41]. The study was conducted twice.

*Statistical Analyses*

One-way Analysis of Variance (ANOVA) was performed on the germination percentage, expressed as the percentage of germinated seeds at the end of the experiment, in order to test differences in terms of response to salt stress between resistant and sensitive weedy rice and cultivated rice varieties. One-way ANOVA was also used to test differences among weedy rice and rice varieties in terms of percentage of germination reduction compared to the germination of the control at each NaCl concentration. Means were separated through the post-hoc test Waller-Duncan ($\alpha \leq 0.05$). The analyses were carried out using the software R version 3.6.1 [42].

For each weedy rice population and rice variety, a separate regression analysis was performed by fitting the following 3-parameters log logistic model to response between seed germination (dependent variable) and the NaCl concentration (independent variable):

$$Y = \frac{d}{1 + \exp[b(\log(x) - \log(e))]} \tag{1}$$

where $Y$ expressed the seed germination percentage, $x$ the NaCl concentration (mM), $d$ the upper limit and $b$ the relative slope at the point of inflection $e$. The regression analysis was performed using the function *drm* of the add-on package *drc* of the software R [42,43]. The same model was used on root and shoot length data ($Y$), transformed as a percentage relative to control (0 mM concentration), as affected by different NaCl concentrations ($x$). Similarly, this regression model was also applied to describe the evolution in time of the germination percentage of each weed population and rice variety, separately for each NaCl concentration. In this case, the parameter $x$ indicated the time expressed in days, while the dependent variable $Y$ expressed the germination percentage.

The function *ED* of the package *drc* was used to calculate the salt concentrations required to reduce by 50% the seed germination, root length and shoot length compared to the values observed at 0mM NaCl ($EC_{50}$), and the germination speed, as the time in days required to reach 50% germination ($T_{50}$) at each NaCl concentration for each population/variety. $EC_{50}$ and $T_{50}$ values were compared among populations/varieties separately for each NaCl concentration using the function *LSD.test* (least significant difference, with Bonferroni adjustment) of the package *agricolae* of the software R.

## 3. Results

### 3.1. Seed Germination Percentage

The results obtained at the end of the trial showed that germination of both weedy rice and cultivated rice was negatively affected by the increase of NaCl concentration (Supplemental Figure S1). The ANOVA among all the weedy rice populations and rice varieties, carried out separately for each NaCl concentration, showed significant differences in germination at 50, 100, 200, 350 and 400 mM NaCl (Table 1). At these salt concentrations, generally, the average germination of sensitive weedy rice populations resulted higher than that of both resistant weedy rice and rice CL80, and ranged from values higher than 90% at 0 mM to values below 15% at 400 mM. However, populations s1 and s3 showed intermediate germination levels at high NaCl concentrations (200 and 350 mM), and were not statistically different from both resistant and sensitive weedy rice and rice varieties. The percentage of germination in the sensitive populations and Baldo variety showed values above 80% at 200 mM NaCl concentration, in some cases above 90%, while the resistant population and CL80 recorded a germination percentage slightly above 50%. Thus, resistant weedy rice behaved similarly to the resistant rice variety CL80, always recording a lower germination level than Baldo and than the sensitive weedy rice populations in most cases.

At the highest NaCl concentration (400 mM), all the populations showed a similar and low germination, while Baldo still showed a germination level above 40%.

In terms of germination reduction, the ANOVA carried out among all of the weedy rice populations and rice varieties showed significant differences at 50, 350 and 400 mM (Table 1).

At 50 mM the germination slightly increased compared to the control (as indicated by negative values in Table 1), showing a hormesis effect of low salt concentration. The highest germination increase was recorded for the resistant weedy rice, with an increase of 17%, followed by the CL80 and s2 population with a germination increase of about 7%. Population s4 was the only population that at 50 mM NaCl did not show hormesis.

At 350 mM, the resistant weedy rice showed the strongest germination reduction, almost 100%, followed by CL80 and s1 that recorded a reduction above 80%. The rice variety Baldo displayed the

lowest germination reduction, almost 35%, while the other sensitive weedy rice populations showed intermediate germination reduction values that ranged from 50% to about 75% reduction.

At the highest salt concentration, the germination reduction was above 90% for the majority of weedy rice populations and for CL80; the Baldo variety recorded the lowest reduction with a value of about 57%, being significantly different to all weedy rice populations and CL80 rice variety.

**Table 1.** Seed germination percentage (after 15 days) and germination reduction at each NaCl concentration compared to control of weedy rice and rice varieties. Comparisons were made among rice varieties and weedy rice populations, separately for each NaCl concentration (among values on the same row). Values sharing the same letters are not significantly different according to Waller-Duncan test ($\alpha \leq 0.05$).

| NaCl concentration (mM) | Germination of weedy rice populations | | | | | Germination of rice varieties | | one-way ANOVA $p$ value |
|---|---|---|---|---|---|---|---|---|
| | s1 | s2 | s3 | s4 | r | Baldo | CL80 | |
| 0 | 93.33 | 91.67 | 91.67 | 96.67 | 76.67 | 96.67 | 88.33 | 0.055 |
| 50 | 98.33b | 98.33b | 96.67b | 96.67b | 90.00a | 99.00b | 94.33ab | 0.027 |
| 100 | 88.33abc | 94.67bc | 98.33c | 96.67c | 80.00a | 98.33c | 85.00ab | 0.012 |
| 150 | 86.67 | 91.67 | 90.00 | 99.33 | 71.67 | 98.33 | 86.67 | 0.074 |
| 200 | 76.67abc | 83.33c | 81.67bc | 91.67c | 53.33a | 95.00c | 58.33ab | 0.010 |
| 250 | 75.00 | 75.00 | 53.33 | 78.33 | 48.33 | 65.00 | 56.67 | 0.084 |
| 300 | 66.67 | 58.33 | 45.00 | 75.00 | 36.67 | 61.67 | 35.00 | 0.084 |
| 350 | 10.00ab | 33.33bc | 23.33abc | 48.33cd | 0.33a | 63.33d | 15.00ab | 0.002 |
| 400 | 6.67a | 15.00a | 1.67a | 10.00a | 1.67a | 41.67b | 3.33a | 0.001 |

| Germination reduction compared to control (%) | | | | | | | | |
|---|---|---|---|---|---|---|---|---|

| NaCl concentration (mM) | Germination reduction of weedy rice populations | | | | | Germination reduction of rice varieties | | one-way ANOVA $p$ value |
|---|---|---|---|---|---|---|---|---|
| | s1 | s2 | s3 | s4 | r | Baldo | CL80 | |
| 50 | −5.36bc [1] | −7.27b | −5.45bc | 0.00c | −17.39a | −2.41bc | −6.79b | 0.010 |
| 100 | 5.36 | −3.27 | −7.27 | 0.00 | −4.35 | −1.72 | 3.78 | 0.334 |
| 150 | 7.14 | 0.00 | 1.82 | −2.76 | 6.52 | −1.72 | 1.89 | 0.940 |
| 200 | 17.86 | 9.09 | 10.91 | 5.17 | 30.43 | 1.72 | 33.96 | 0.092 |
| 250 | 19.64 | 18.18 | 41.82 | 18.96 | 36.96 | 32.76 | 35.85 | 0.342 |
| 300 | 28.57 | 36.36 | 50.91 | 22.43 | 52.17 | 36.21 | 60.38 | 0.454 |
| 350 | 89.28cd | 63.64bc | 74.54bcd | 50.00ab | 99.56d | 34.48a | 83.02cd | 0.003 |
| 400 | 92.86b | 83.64b | 98.18b | 89.65b | 97.83b | 56.90a | 96.23b | 0.001 |

[1] Negative values represent an increase of germination.

The regression analysis carried out between germination percentage and NaCl concentrations also highlighted the decreasing trend of the germination percentage at increasing salt concentrations for each population, with the variety Baldo showing the smallest decrease (Supplemental Figure S1).

The post-hoc analysis carried out on $EC_{50}$ among weedy rice and rice varieties showed that both resistant rice and weedy rice required significantly lower NaCl concentrations (lower than 250 mM) compared to the sensitive ones to reduce their germination by 50% (Table 2). Population s3 showed an intermediate behaviour between resistant and sensitive populations/varieties, necessitating a salt concentration of about 270 mM to reduce their germination by 50%. Baldo confirmed to be the most tolerant to salt, showing an $EC_{50}$ of about 370 mM NaCl (Table 2).

**Table 2.** NaCl concentrations (mM) required to reduce by 50% seed germination (EC$_{50}$ ± SE) of the tested weedy rice populations and rice varieties. Values sharing the same letters are not significantly different according to a least significant distance (LSD) test with Bonferroni adjustment ($\alpha \leq 0.05$).

| Weedy Rice Population/Rice Variety | EC$_{50}$ |
|---|---|
| s1 | 312.80 ± 6.06 c |
| s2 | 310.32 ± 11.27 bc |
| s3 | 270.16 ± 11.06 ab |
| s4 | 338.35 ± 9.47 cd |
| r | 229.70 ± 17.11 a |
| Baldo | 369.63 ± 22.67 d |
| CL80 | 247.27 ± 13.72 a |

## 3.2. Germination Speed

Salinity influenced not only the germination level of seeds, but also the time required to reach a same level of germination in both weedy rice and rice varieties. The time required to reach 50% germination showed a difference between weedy rice populations and cultivated rice varieties already at 0 mM, regardless of their herbicide sensitivity (Table 3). At this concentration, all rice varieties required less than two days to reach 50% germination, while all the weedy rice populations required about three days to reach the same level of germination; the only exception was the resistant population r which, similarly to Baldo, also required about two days to reach 50% germination. In general, Baldo at all of the tested concentrations was the fastest to reach 50% germination. Resistant rice and weedy rice had a similar germination speed, which was generally intermediate between that of Baldo and the sensitive weedy rice. Sensitive weedy rice populations showed in general the highest T$_{50}$ values, and only at concentrations higher than 150 mM NaCl some sensitive populations showed a T$_{50}$ not statistically different from Baldo. Resistant weedy rice did not reach 50% germination starting from 250 mM. At the highest NaCl concentrations (350 and 400 mM) the majority of the populations did not reach a germination of 50%; the only exception was Baldo at 350 mM that had a T$_{50}$ of about 6.8 days.

**Table 3.** Time (days) required to reach 50% seed germination (T$_{50}$ ± SE) of the tested weedy rice populations and rice varieties at each NaCl concentration. Comparisons were made among rice varieties and weedy rice populations, separately for each NaCl concentration (among values on the same row). Values sharing the same letters are not significantly different according to the LSD test with Bonferroni adjustment ($\alpha \leq 0.05$).

| NaCl Concentration (mM) | Weedy Rice Population/Rice Variety | | | | | | |
|---|---|---|---|---|---|---|---|
| | s1 | s2 | s3 | s4 | r | Baldo | CL80 |
| 0 | 2.71 ± 0.06 b | 3.82 ± 0.09 c | 3.44 ± 0.08 c | 3.20 ± 0.08 bc | 1.87 ± 0.11 a | 1.73 ± 0.55 a | 1.57 ± 0.08 a |
| 50 | 3.06 ±0.06 cd | 3.81 ± 0.06 e | 3.23 ± 0.06 d | 2.88 ± 0.04 c | 2.05 ± 0.07 b | 1.69 ± 0.12 a | 2.06 ± 0.07 b |
| 100 | 3.20 ± 0.07 d | 4.24 ± 0.08 f | 3.60 ± 0.07 e | 3.48 ± 0.08 e | 2.59 ± 0.12 c | 2.08 ± 0.04 a | 2.34 ± 0.09 b |
| 150 | 4.29 ± 0.10 c | 4.76 ± 0.12 de | 5.03 ± 0.13 e | 4.38 ± 0.11 cd | 4.19 ± 0.26 c | 2.47 ± 0.08 a | 3.18 ± 0.13 b |
| 200 | 6.17 ± 0.21 bc | 6.04 ± 0.23 bc | 6.46 ± 0.31 c | 5.73 ± 0.30 ab | 10.16 ± 2.78 c | 3.52 ± 0.18 a | 8.84 ± 1.32 bc |
| 250 | 7.78 ± 0.31 ab | 7.32 ± 0.32 ab | 10.52 ± 1.57 b | 8.36 ± 1.16 ab | - | 4.78 ± 0.55 a | 10.14 ± 2.68 b |
| 300 | 11.25 ± 0.61 ab | 9.60 ± 0.66 a | - | 10.25 ± 1.07 ab | - | 9.01 ± 2.04 a | - |
| 350 | - | - | - | - | - | 6.78±0.66 | - |
| 400 | - | - | - | - | - | - | - |

## 3.3. Root and Shoot Length

Different salinity conditions were able not only to affect germination percentage and speed, but also to reduce root and shoot growth measured at 15 days after the beginning of the germination test. In particular, a decrease of root and shoot development with increasing salt concentrations was observed (Figure 1).

Root length varied on average, for sensitive weedy rice populations, from about 10 cm at 0 mM NaCl to 0.25 cm at 350 mM NaCl (less than 5% of the length of the control; Figure 1A). These populations did not produce roots at higher concentrations. Baldo was the only variety able to develop roots at 400 mM NaCl.

In the resistant weedy rice population the root length varied from about 7.40 cm at 0 mM to 0.72 cm at 300 mM, and no root development was observed at concentrations higher than 300 mM; however the root length at this concentration still represented 9.7% of the control length.

In the case of rice varieties, for Baldo root growth ranged from about 7.90 cm at 0 mM NaCl to 0.54 cm at 350 NaCl (representing about 7% of the length of the control) and for CL80 between about 4.40 cm at 0 mM and 0.50 cm at 350 mM NaCl (about 10% of the control length); in the second variety no root development was recorded at 400 mM NaCl.

Figure 1A showed that root length reduction at the 350 mM salt concentration compared to the control was above 90% for all the weedy rice populations and rice varieties.

Sensitive weedy rice populations showed the NaCl concentration required to reduce root length by 50% significantly lower than rice varieties, with values ranging from about 140.7 mM (s1) to 173.4 mM (s3 population) (Table 4). Resistant weedy rice populations showed an intermediate behaviour between sensitive population and rice varieties, requiring NaCl concentrations slightly above 200 mM to reduce its root length by 50%. A higher root salinity tolerance was oberved in both rice varieties, with CL80 being the one that necessitated a higher salt concentration to reduce root length by the same level.

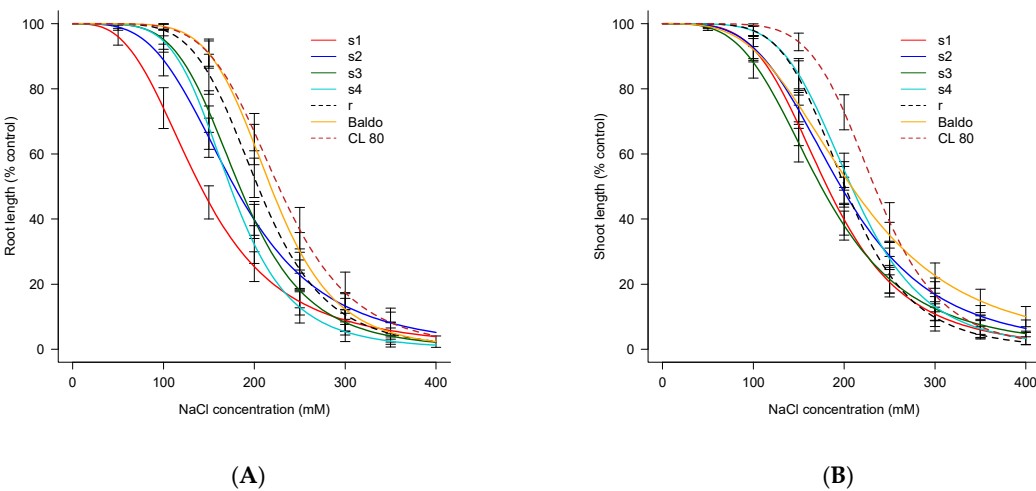

**(A)** **(B)**

**Figure 1.** Root length (**A**) and shoot length (**B**) of the weedy rice populations and the rice varieties at the different NaCl concentrations as percentage relative to control (0 mM NaCl concentration).

Shoot development decreased as salt concentration increased (Figure 1B). The shoot length of the tested weedy rice populations ranged, on average, from 6.70 cm at 0 mM to 0.18 cm at 400 mM (less than 6% of the shoot length recorded in the control). In general, sensitive populations showed a higher shoot development than the resistant one; in fact, population r was able to produce shoots only up to 300 mM NaCl, while CL80 only up to 350 mM. At 300 mM, shoot length reduction of resistant population was similar, or sometimes lower, than sensitive populations.

In rice varieties, shoot length ranged on average, from 6.46 cm at 0 mM to 0.61 cm at 400 mM. At all NaCl concentrations, Baldo produced longer shoots than CL80, and at 350 mM its shoot length was slightly less reduced compared to CL80 (86% reduction for Baldo compared to 91% for CL80).

The majority of sensitive weedy rice populations required a lower NaCl concentration to reduce their shoot length by 50% compared to rice varieties (Table 4). Resistant weedy rice had an intermediate behaviour between weedy rice and rice, while the sensitive population s4 behaved similarly to Baldo, and required a higher NaCl concentration to reduce its shoot length (about 207 mM NaCl). As for root,

CL80 required higher salt concentration to reduce its shoot length compared to all the other rice and weedy rice.

**Table 4.** NaCl concentrations (mM) required to reduce by 50% root and shoot length ($EC_{50} \pm SE$) of the tested weedy rice populations and rice varieties. Values sharing the same letters are not significantly different according to the LSD test with Bonferroni adjustment ($\alpha \leq 0.05$).

| Weedy Rice Population/Rice Variety | $EC_{50}$ Root | $EC_{50}$ Shoot |
|:---:|:---:|:---:|
| s1 | 140.74 ± 4.99 a | 181.13 ± 4.51 b |
| s2 | 178.13 ± 5.20 c | 196.30 ± 4.97 c |
| s3 | 183.71 ± 4.43 d | 174.54 ± 4.90 a |
| s4 | 173.40 ± 4.11 b | 207.89 ± 4.21 e |
| r | 203.88 ± 4.51 e | 200.72 ± 4.18 d |
| Baldo | 217.92 ± 4.22 f | 207.93 ± 5.37 e |
| CL80 | 226.62 ± 5.26 g | 233.30 ± 4.29 f |

## 4. Discussion

The results of the study highlighted the influence of saline concentrations on the germination of both resistant and sensitive rice and weedy rice. At the lowest tested NaCl concentrations (50–100 mM), the two rice varieties and the majority of weedy rice populations showed a higher germination compared to control. This phenomenon can be attributed to a hormetic effect of salt, in which low doses can stimulate germination, while inhibition is commonly observed at higher doses [44]. This higher germination was more evident in the resistant rice (CL80) and in the resistant weedy rice population. A certain level of hormesis due to salt was demonstrated in a previous study in which weed germination was assessed from a seedbank of coastal saline soil from China. In this study, in the soil seedbank having the lowest level of salinity (from 0 to 800 mg salt $kg^{-1}$ soil), the germination of different plant species increased with increasing levels of salt up to 800 mg $kg^{-1}$. An opposite behaviour was instead observed with soils having a salt concentration above 800 mg $kg^{-1}$ [45]. The hormetic effect observed in our study was more pronounced in resistant rice and weedy rice, thus in the case of moderate salinity level the cultivation of this Clearfield® variety can be possible, without reducing germination. However, the imazamox-resistant weedy rice tested in our study was even more stimulated in the germination by the low rate of NaCl. Even though our study was conducted on a single resistant weedy rice population, the fact that the behaviour of imidazolinone-resistant rice and weedy rice was similar suggests the existence of a different response to salinity between sensitive and resistant *Oryza sativa*. A soil salinity concentration of 40 mM NaCl has been found to be a moderate salinity level in which rice can still grow, and 60 mM NaCl is the threshold value at which rice shows symptoms due to salinity [36,46]. At these salinity levels, germination stimulation caused by hormesis may represent an additional issue of the selection of resistant weedy rice populations.

In our study, at NaCl concentrations above 200 mM, resistant weedy rice and CL80 showed significantly lower germination with the largest reduction compared to the control without NaCl than the sensitive weedy rice populations and Baldo. This different behaviour was maintained up to the highest salinity levels.

In areas with saline problems, the cultivation of rice varieties tolerant to salt can still guarantee a high germination and seedling growth, which are the key stages for a good crop establishment and high competitiveness towards weeds [27,47].

Moreover, as the early crop growth stages are the periods in which plants are more sensitive to abiotic stresses, salt tolerance in this phase in saline environments can help the crop to cope with these stresses [48]. The availability of rice varieties tolerant to salinity is particularly important to maintain high yield, considering that rice is one of the most sensitive crops to salt, and that the majority of the rice areas worldwide are located near river deltas and in coastal areas that have saltwater intrusion problems [36,49]. Other studies have demonstrated that Baldo has a higher salt tolerance at seedling

stage due to its ability to early activate the response to salinity by reducing salt uptake by its roots and by compartmenting ions in the roots and cell vacuole [36,50]. Nevertheless, Baldo has been classified either as sensitive [50] or salt tolerant at the seedling stage [36], according to the relative salt sensitivity of the varieties to which it was compared. However, salt tolerance of rice varieties and weedy rice during germination has been less studied. The comparison of the germination time of weeds and crop is a key element in the definition of competition in certain environments.

The imazamox-resistant weedy rice population tested in this study appeared to suffer for salt stress more than the other sensitive weedy rice populations in terms of germination reduction. In case of high salinity levels, the resistant weedy rice could potentially be less problematic as its germination is suppressed. This may be partially compensated by a faster germination because resistant weedy rice was able to reach 50% germination in a shorter time than the sensitive populations at NaCl concentrations up to 100 mM. Clearfield® rice germination has also been inhibited in the saline environment. Salt tolerance and its correlation with herbicide resistance in both rice and weedy rice has not been investigated up to now. Weedy rice resistant to herbicides can have an ecological advantage or disadvantage over the other populations. For example, resistant populations can be more tolerant to abiotic stresses, as it has been demonstrated for some weedy rice hybrids derived from reverse gene flow from weedy rice to cultivated rice [13]. Also, a higher germination speed has been found in herbicide-resistant weeds having an ALS gene mutation, that results in a faster spread of the resistance [51]. The same observation can be done for the resistant weedy rice population of the present study; in fact, at 0 mM the germination speed was higher than that of the sensitive weedy rice populations, even though it was similar to that of both rice varieties.

Gene flow from resistant crops, as well as spontaneous mutations conferring resistance to herbicides, can induce in some resistant populations a fitness penalty [51]. In our study, the weedy rice resistant population, despite having a higher speed of germination, showed a lower germination percentage at higher saline concentrations. The same disadvantage was observed in the resistant rice CL80. This behaviour could be due to several factors, including the result of a different genetic background, as well as to a fitness penalty in imidazolinone-resistant *O. sativa*. This could be clarified only through specific studies in which several herbicide-susceptible and resistant rice varieties and weedy rice populations are compared.

Weedy rice populations showed a variable germination behaviour, not only between resistant and sensitive, but also among sensitive populations. A variable germination among weedy rice populations was found in previous studies even in non-saline conditions, and a correlation with other morphological traits (hull colouration, awn presence) has been demonstrated [4,16,52]. The variable response to salt has already been shown for other weed species, such as *E. crus-galli*, and it can be considered as an advantage for weeds that can easily adapt and infest rice in different environments [38,53].

Root and shoot length were deeply influenced by salinity as well. As observed in other studies [34] a decrease in root and shoot length with increasing salt concentrations was observed. In our study, rice varieties showed a lower sensitivity to NaCl in terms of root and shoot reduction. The $EC_{50}$ for both root and shoot length of the two tested rice varieties was higher than the values observed for weedy rice populations, especially for sensitive ones. This different behaviour may be relevant in case of rice cultivation in salt-stress conditions, since weeds seem to be more susceptible to NaCl in terms of root and shoot length reduction than rice, and they may compete less towards the crop. CL80 showed the highest tolerance to salt, followed by Baldo; however, only Baldo was still able to produce roots at 400 mM NaCl.

It has been observed that Baldo concentrates salt more in the roots compared to shoots, and this confers to the variety a higher salt tolerance because roots are generally more salt tolerant than shoots [36]. In fact, Baldo also showed higher shoot length than CL80, but a lower concentration of salt was necessary to reduce shoot length by 50%, highlighting a higher sensitivity of the shoots compared to roots.

The resistant weedy rice population required a higher salt concentration to reduce 50% root length compared to all the sensitive weedy rice, even though it was the only population not able to produce roots at concentrations higher than 300 mM. Salt concentrations required to reduce shoot length of the r population by 50% were higher than the majority of weedy rice populations, but lower than population s4 and Baldo.

The results of the study highlighted a common behaviour of resistant weedy rice and rice (r and CL80) for the majority of the germination and growth parameters measured. In particular, they suffered more in case of high salt concentration, as their germination was highly reduced. However, this can be partially compensated by their faster germination compared to sensitive weedy rice populations, and by the higher salt concentration required to reduce root and shoot length, in particular in the case of CL80. Thus, in case of a saline environment, resistant populations can be disfavoured, because of their lower germination and their inability to produce roots and shoots at a high salt level. Baldo was the most tolerant to salt compared to both rice and weedy rice, and represents a variety that can be cultivated even in saline environments.

As soil salinisation and salt-water intrusion are becoming more and more frequent in many areas of rice cultivation, the knowledge of the ability of both weeds and crops to tolerate this stress is particularly important, and it has consequences on weed control [18]. Further studies are necessary to clarify the correlation between salt tolerance and herbicide sensitivity, even at more advanced growth stages.

**Supplementary Materials:** The following are available online at http://www.mdpi.com/2073-4395/9/10/658/s1, Figure S1: Germination percentage of the tested weedy rice populations and rice varieties as a function of NaCl concentration, as fitted by the following log-logistic model $Y = d/\{1 + \exp[b(\log(x) - \log(e))]\}$.

**Author Contributions:** Conceptualisation, S.F., F.S. and F.V.; methodology, F.S.; data curation, S.F., F.S., M.M. and F.V.; writing—original draft preparation, F.S. and L.P.; writing—review and editing, S.F., M.M. and F.V.; supervision, F.V.

**Funding:** This research received no external funding.

**Conflicts of Interest:** The authors declare no conflict of interest.

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
