# Peer review of "Effect of Different Water Salinity Levels on the Germination of Imazamox-Resistant and Sensitive Weedy Rice and Cultivated Rice"

_agronomy, doi:10.3390/agronomy9100658_

Round 1

Reviewer 1 Report

General comments: the authors have done an excellent job of revising their manuscript. I have a few minor suggestions for further improvements:

(1) Line 18 and throughout manuscript: there is a lot of disagreement between researchers over whether “germination rate” refers to the final germination percentage or the speed at which germination occurs. Therefore, I suggest that “germination rate” be replaced with “germination percentage” throughout the manuscript, so that some readers will not be confused by thinking that the authors’ measurements of germination “rate” and “speed” refer to the same thing.

(2) Lines 36, 72 and 78: replace “damages” with “damage”.

(3) Line 44: this sentence can be simplified to “...imazamox were identified only four years after the introduction of Clearfield to Italy, which occurred in 2006.”

(4) Lines 51 – 52 and 104: be consistent as to whether the Italian or English versions of place names are used.

(5) Line 77: replace “instead responsible of” with “also responsible for”.

(6) Line 106: replace “The resistant of r population” with “The resistance of the r population”.

(7) Line 114: place this green-highlighted sentence at the end of the previous paragraph, and start the new paragraph with the “Weedy rice seeds...” sentence.

(8) Line 151: should this variable Y be germination speed?

(9) Line 154: wasn’t the calculation for germination speed just described in the previous paragraph?

(10) Line 155: delete “rate” because the phrase “required to reach 50% germination” is perfectly adequate.

(11) Line 202: should the question mark after “Supplementary figure” be replaced with a number?

(12) Line 222: although it is stated that all the weedy rice populations required 3 days to reach 50% germination, the r population only required 2 days and was not significantly different to Baldo and CL80 at 0 mM NaCl. This should be mentioned.

(13) Line 227: the paragraph would be clearer if the sentence about “...only at higher concentrations than 150 mM NaCl some sensitive populations...” on lines 223 – 225 was placed after the sentence ending “...the highest T50 values” on line 227.

(14) Line 252: replace “lenght” with “length”.

(15) Line 311: it would be better to delete “such as Baldo” and start this part of the Discussion by referring to salt-tolerant rice varieties in general. Then the sentence about Baldo on lines 321- 323 will make more sense.

(16) Lines 328 – 329: I suggest toning down the language slightly by replacing the “can” on line 328 with “could potentially” and the “can” on line 329 with “may”.

(17) Line 346: delete “Nevertheless” and start the sentence with “This”.

(18) Line 371: replace “off” with “of”.

(19) Line 381: replace “resulted” with “represents”.

Reviewer 2 Report

I have gone through the comments from the authors and I am satisfied with the corrections.

Author Response

No further corrections were required by the reviewer

This manuscript is a resubmission of an earlier submission. The following is a list of the peer review reports and author responses from that submission.

Round 1

Reviewer 1 Report

13:  aim instead of aims

18:  I recommend "rate of germination" instead of germination speed (change throughout manuscript)

43:  delete "In Italy, in particular"  Several weedy rice.....

50-51:  awn presence, or variability in physiological traits including

62:  In Italy where a majority of the European rice production occurs, this phenomenon....

72:  In rice, previous studies have shown that

75:  among different rice varieties[28,29].

75-77:  Reword the sentence "The salinity level tolerated by rice during germination seems not correlated with salinity tolerance at other growth stage; a previous study found that only few varieties were slightly affected by a salinity level of 12 dS m-1 [29,30]".  I would break into two sentences to start.

89:  imazamox were investigated

95-96:  collected in 2008-2009.

98:  s4) were sensitive to label rate of imazamox while one (labelled r) was resistant.

109:  recorded in selected European rice cultivation areas [17,32-34].

112:  experimental unit as a single Petri

154:  (200 and 350 mM) and were not statistically different

198:  see comment from line 18

199:  but also the time required for germination in

Figure 1,2,3,4:  I think it would help to have the colored lines next to the s1, s2, s3, s4, r, Baldo, and CL80 in the caption.

Table 2:  What does SI values mean in the table caption?

236:  population, the root length varied...

242:  CL80; however, Baldo had developed roots at 400 mM

297-299:  Delete paragraph, a restatement of the objectives.

300:  saline concentrations on the germination of both

304:  doses can stimulate germination while inhibition

306-310:  Sentence as written is way to long (run-on).  This needs to split into at least 3 separate sentences.

311: effect observed in our study

312-313:  possible without reducing germination.

314:  germination by the low rate of NaCl.

321:  At NaCl concentrations above 200 mM, resistant weedy rice

322-324:  lower germination with the largest reduction compared to the control without NaCl than the sensitive weedy rice populations and Baldo.  This different behaviour was observed at the highest salinity levels.

327:  The early crop growth

332:  that have saltwater intrusion

333:  stage due to its ability

334:  delete "early"

370:  salt-stress conditions since weeds

Reviewer 2 Report

General comments: This is an interesting manuscript on the relative salinity tolerance of IMI herbicide-susceptible and -resistant rice at the stage of seed germination and early seedling growth. The experiments were generally well-designed, although a salt-susceptible, IMI-susceptible control rice cultivar is missing. The presentation of the results is currently somewhat confusing and needs to be improved (see specific comments). Overall, the conclusions are sound but some are unsupported by experimental evidence or are too broad.

Specific comments:

Abstract

(1) Line 11: replace “threat” with “threaten”.

(2) Line 12: replace “costal” with “coastal”.

(3) Lines 22 – 23: the information about root and shoot growth appears to be added as an afterthought rather than as important data. The fact that root and shoot growth were measured needs to be added earlier, for instance in the description of the experiment on line 13, i.e. “…the effects of salinity on germination and early seedling growth of Italian weedy rice…” The results of the root and shoot measurements need to be stated in more detail, as was done for the germination data.

Introduction

(4) Line 44: please state the year that Clearfield crops were introduced to Italy.

(5) Lines 70 – 71: does salinity actually induce seed dormancy, or merely inhibit seed germination?

(6) Line 77: the wording of this sentence makes it unclear whether the varieties in question were affected by salinity at the germination stage or at post-germination stages; and the two references cited deal with germination and early seedling growth, respectively, so it remains unclear.

(7) Line 84: the concept of herbicide resistance is introduced very suddenly here, after so much previous discussion of salinity. It might be worth adding a short paragraph on the nature and extent of the herbicide resistance problem in Italian rice fields so that the reader has some context for why this study was performed.

Materials and methods

(8) Line 98: is the IMI resistance mechanism of weedy rice population “r” known? An ALS mutation vs. enhanced herbicide metabolism might have implications for the level of general abiotic stress tolerance of the population.

(9) Line 99: it is stated here that Baldo is a conventional rice variety, but late in the Discussion the fact that Baldo is relatively salt-tolerant is mentioned for the first time. Therefore, it is not really appropriate to compare Baldo to CL80 without also including a rice variety that is known to be susceptible to both salt and IMI herbicide (Vialone Nano seems to be commonly used as a salt-sensitive cultivar in the literature). If it is not possible to include such a cultivar in the study, the authors might consider omitting the rice cultivars and focusing solely on the weedy rice populations.

(10) Line 101: is the initial seed viability of each rice population/cultivar known? If it is less than 100%, the data needs to be corrected for the percentage of seeds that cannot germinate under any conditions.

(11) Lines 138 – 139: the concept of “time required to reduce by 50% germination” is very difficult to understand, especially since it is defined as being different from the conventional measure of time to 50% germination (T50) in one part of the manuscript(lines 208 – 213), but is then defined as being “time required to reach 50% germination” (i.e. T50) in another part (line 227). It might be a good idea for the authors to stick to the conventional T50 equation (Coolbear et al. 1984: J Exp Bot 35: 1609 – 1617) rather than using the dose-response function in R. (Since seed viability is presumably close to 100% and germination is greatly inhibited at >300 mM NaCl, it would be better to take “50% germination” as 50% of the starting seed population (i.e. germination of 10 seeds) rather than 50% of the final number of germinated seeds.)

Results

(12) Overall, the way the data is presented is quite awkward and difficult to follow. The modelled dose-response curves are not very informative and the use of seven different-coloured solid lines for the seven populations, without a key on the graph, requires a lot of effort for the reader to understand. It would preferable if the authors could plot their actual data, with standard error bars, to show the effect of increasing NaCl concentration on final seed germination percentage (Fig. 1) and on root and shoot growth (expressed as a % of control values) (Fig. 2A and B). These graphs should have different symbols for the data points and solid or dashed lines, as well as different colours, and should include a symbol key rather than making the reader constantly refer to the figure legend to determine which line is which. If the seed germination data was plotted as suggested, the current Table 1 would then be unnecessary, but the authors could include the ANOVA results as a supplementary table if desired. A single table showing the EC50 values for seed germination, root length and shoot length in response to NaCl concentration (new Table 1) would be better than lining up the EC50 values beside each graph, and could include standard errors and indicators of significant differences between populations. It would be unnecessary to include the SI data in the current Table 3, which is quite confusing to read, because the reader could readily compare the EC50 values of the various populations using the new Table 1. It is unnecessary to include the modelled time-course of germination at each NaCl concentration, as these graphs could be replaced by a table (new Table 2) showing the T50 for each population at each NaCl concentration, with standard errors and indicators of significant differences between populations and between treatments within each population. This is a more complete way of treating the data than the current Table 2 and current Fig. 2.

(13) Lines 149 – 150: the wording here makes it unclear whether there were significant differences between populations at each NaCl concentration, or within each population across the different treatments.

(14) Lines 177 – 178: the heading of the current Table 1 needs an explanation of what the letters after the numbers mean (obviously significant differences, but are the comparisons being made down the columns, across the rows, or both?).

(15) Line 200: replace “begin” with “beginning”.

(16) Lines 208 – 213: what was the time to 50% germination under control conditions? Again, the “EC50” for germination time is difficult to comprehend.

(17) Line 220: it is not at all clear what “the time required to reduce by 50% seed germination capacity” actually means.

(18) Line 234: it might be useful to remind the reader here that the root and shoot lengths were measured at 15 d after the start of imbibition.

(19) Lines 240 – 243: since CL80 did not grow roots at all at 400 mM, Baldo is clearly more tolerant, so to say that Baldo experienced a greater reduction in root growth (by comparing its performance at 400 mM with the performance of CL80 at 350 mM) is misleading. The two varieties should be compared at the same NaCl concentration.

(20) Lines 276 – 279: the various EC50 values for shoot growth are stated in the text, but not whether the value for the r population is significantly different from the values for the s populations. The reader has to refer to the awkwardly-presented SI values in the current Table 3. As mentioned above, a table with all EC50 values placed together would be very useful.

Discussion

(21) Line 307: replace “costal” with “coastal”.

(22) Line 325: this is far too late to reveal that Baldo is in fact a known salt-tolerant cultivar.

(23) Lines 332 – 333: the previous characterisation of Baldo as being salt-tolerant at the seedling stage should have been mentioned in the Introduction and/or Methods.

(24) The paragraph beginning line 338 seems to contradict itself. The authors start by stating that the IMI-resistant weedy rice is more sensitive to salt stress, and end by saying that its germination rate was faster in the presence of salt. However, since this faster rate was acknowledged to not always be statistically significant, it might not be worth mentioning at all. The addition of Baldo to the end of the paragraph increases the confusion; a re-wording of the entire paragraph is suggested.

(25) Lines 355 – 357: this study was not a formal fitness study, so the statement that the r population “probably showed a fitness penalty” is weak and unsupported by data. This population presumably has a different genetic background to the s populations, so its higher salt sensitivity could be caused by many other factors. Similarly, it cannot be said that CL80 has a fitness penalty, because it is being compared to a cultivar that is well known to be salt-tolerant. CL80 needs to be compared to several other herbicide-susceptible rice cultivars before conclusions such as this can be drawn.

(26) Line 359: replace “within” with “among”.